# Intragenic Antimicrobial Peptide Hs02 Hampers the Proliferation of Single- and Dual-Species Biofilms of *P. aeruginosa* and *S. aureus*: A Promising Agent for Mitigation of Biofilm-Associated Infections

**DOI:** 10.3390/ijms20143604

**Published:** 2019-07-23

**Authors:** Lucinda J. Bessa, Julia R. Manickchand, Peter Eaton, José Roberto S. A. Leite, Guilherme D. Brand, Paula Gameiro

**Affiliations:** 1LAQV/Requimte, Departamento de Química e Bioquímica, Faculdade de Ciências da, Universidade do Porto, 4050-313 Porto, Portugal; 2Laboratório de Síntese e Análise de Biomoléculas, Instituto de Química, Universidade de Brasília, UnB, Brasília DF 70910-900, Brasil; 3Núcleo de Pesquisa em Morfologia e Imunonologia Aplicada, NuPMIA, Área de Morfologia, Faculdade de Medicina, FM, Universidade de Brasília, UnB, Brasília DF 70910-900, Brasil

**Keywords:** polymicrobial biofilms, intragenic antimicrobial peptide, Hs02, *Pseudomonas aeruginosa*, *Staphylococcus aureus*

## Abstract

*Pseudomonas aeruginosa* and *Staphylococcus aureus* are two major pathogens involved in a large variety of infections. Their co-occurrence in the same site of infection has been frequently reported and is linked to enhanced virulence and difficulty of treatment. Herein, the antimicrobial and antibiofilm activities of an intragenic antimicrobial peptide (IAP), named Hs02, which was uncovered from the human unconventional myosin 1H protein, were investigated against several *P. aeruginosa* and *S. aureus* strains, including multidrug-resistant (MDR) isolates. The antibiofilm activity was evaluated on single- and dual-species biofilms of *P. aeruginosa* and *S. aureus*. Moreover, the effect of peptide Hs02 on the membrane fluidity of the strains was assessed through Laurdan generalized polarization (GP). Minimum inhibitory concentration (MIC) values of peptide Hs02 ranged from 2 to 16 μg/mL against all strains and MDR isolates. Though Hs02 was not able to hamper biofilm formation by some strains at sub-MIC values, it clearly affected 24 h preformed biofilms, especially by reducing the viability of the bacterial cells within the single- and dual-species biofilms, as shown by confocal laser scanning microscopy (CLSM) and atomic force microscopy (AFM) images. Laurdan GP values showed that Hs02 induces membrane rigidification in both *P. aeruginosa* and *S. aureus*. Peptide Hs02 can potentially be a lead for further improvement as an antibiofilm agent.

## 1. Introduction

Most types of infections, especially chronic infections, are of polymicrobial origin; that is, two or more microorganisms are present and play a role themselves in the infection [1,2,3]. Within the polymicrobial context, various bacterial species communicate, cooperate, and compete with each other [4]. Polymicrobial infections consisting of two or more bacterial pathogens are common in wounds and in cystic fibrosis lung infections and usually are biofilm-associated, which can greatly hamper the efficacy of treatment and thus of healing [5,6,7].

*Staphylococcus aureus* and *Pseudomonas aeruginosa* are two important bacterial pathogens that are known to have developed complex interactions in such chronic polymicrobial infections [4,8,9,10]. Though an antagonistic relationship was initially considered to occur between these two species, recently, *P. aeruginosa* and *S. aureus* have been isolated from the same site of infection, with evidence that both pathogens seem to worsen the infection’s evolution [11,12,13]. According to Hotterbeekx et al. [4], there are several studies demonstrating that the copresence of *P. aeruginosa* and *S. aureus* in vivo is linked to worse disease outcomes and to a delay in the healing process.

Interspecies interactions differ between the planktonic and biofilm modes of growth [4,14]. Bacteria in biofilms exhibit different gene expression, growth rates, behaviors, and appearances to those that are in the planktonic state [15], and it is recognized that biofilms are more tolerant to antibiotics than planktonic cells [16]. Moreover, *S. aureus* interaction with *P. aeruginosa* within a biofilm can alter *S. aureus*’ susceptibility to different antibiotics [9,17]. 

Antimicrobial peptides (AMPs) have been isolated from a variety of organisms, such as bacteria, reptiles, plants, and mammals and can also be chemically synthesized [18]. AMPs are regarded as promising alternatives to conventional antibiotics [18,19], presenting several mechanisms of action, and usually, more than one mechanism is present simultaneously [20,21]. There are also AMPs with the ability to specifically affect biofilms by inhibiting the biofilm formation or killing preformed biofilms [19,22,23]. It has been demonstrated that proteins from varied organisms contain encrypted structural elements with significant physicochemical similarity to AMPs, termed intragenic antimicrobial peptides (IAPs) [24,25]. This opens an exciting opportunity to screen organism genomes for encrypted bioactive molecules with therapeutic potential. Once identified and synthesized as individual entities, IAPs may share biological activities with AMPs, such as broad and direct antimicrobial activity against susceptible and resistant bacterial strains, capacity to act as anti-inflammatory agents, to exert chemoattractant effect in immune cells, and also to disrupt the formation of biofilms [10,22,26]. The peptide Hs02 was identified as an internal fragment of the unconventional myosin 1H protein from a collection of human proteins using the software Kamal, which was originally developed to probe IAPs in plant proteins [24,25,27]. Hs02 was demonstrated to fold as an amphiphilic α-helix upon membrane interaction and to present potent antimicrobial and anti-inflammatory activity [27]. On that basis, in this study, it was aimed to evaluate the antibiofilm activity of the peptide against *P. aeruginosa* and *S. aureus* single- and dual-species biofilms formed by multidrug-resistant isolates.

## 2. Results and Discussion

### 2.1. Antibacterial Activity of Peptide Hs02

Peptide Hs02 exhibited antibacterial activity against both Gram-positive and Gram-negative strains, including multidrug-resistant clinical isolates (Table 1). Minimum inhibitory concentration (MIC) and minimum bactericidal concentration (MBC) values ranged from 2 to 16 μg/mL (1 to 8.2 µM), which are low values and thus indicative of effective antimicrobial activity. These results confirm the previously reported antimicrobial activity of the peptide [27]. Like many other AMPs, peptide Hs02 is bactericidal and has a broad-spectrum action [28].

### 2.2. Antibiofilm Activity of Peptide Hs02

#### 2.2.1. Peptide Hs02 Did Not Inhibit Biofilm Formation

No biofilm was formed by any of the isolates in the presence of the peptide at its MIC, as expected. Nonetheless, peptide Hs02 did not inhibit biofilm formation by the PA004, PA008, and Sa1 isolates in presence of subinhibitory concentrations, since the biofilm formed was similar or greater than the control biofilm (grown in the absence of peptide Hs02) (Figure 1). For the isolate SA007, however, at the sub-MIC levels of 0.5× and 0.25× MIC, the biofilm formed was significantly reduced in comparison to the control.

#### 2.2.2. Peptide Hs02 Hampered the Proliferation of 24 h Biofilms

Peptide Hs02 had a more marked effect on 24 h performed biofilms (either single- and dual-species), by decreasing their proliferation at concentrations of 8× MIC. The biofilm proliferation was recorded by measuring the optical density at 600 nm (OD_600_) of the planktonic phase of the treated and nontreated biofilms after 24 h (Figure 2). The single-species biofilms of *P. aeruginosa* isolates PA002 and Pa3 were less affected by peptide Hs02 than those of *S. aureus*, which were clearly affected for all three strains—SA007, Sa3, and Sa1. Two out of the three dual-species biofilms (PA002+Sa3 and Pa3+Sa1) suffered an intermediate reduction when treated with the peptide.

#### 2.2.3. Peptide Hs02-Treated Biofilms Presented Reduced Viability

By using confocal laser scanning microscopy (CLSM), it could be observed that the peptide-treated biofilms maintained overall structure in comparison to the respective nontreated ones; however, the viable cells within the treated biofilms were reduced to some extent in all cases (Figure 3). As it is generally accepted that AMPs interact with the cytoplasmic membrane, causing membrane rupture that will eventually lead to cell lysis [29], one could expect to see more red (nonviable) cells in the peptide Hs02-treated biofilms, as was indeed observed. The peptide effect was quite striking in the case of both *P. aeruginosa* biofilms assayed, Pa4 and PA002, where more than 90% of the cells within the biofilm were red.

The dual-species biofilm of PA002+Sa3 was also analyzed by atomic force microscopy (AFM) (Figure 4). As we can see, the peptide-treated biofilm presents some undamaged cells, but clearly more damaged ones (“dead cells”) in comparison to the nontreated biofilm. Again, we saw more damaged PA002 cells (in the form of empty membranes or “ghosts”) than Sa3 cells, which in general looked quite intact in the AFM images. It is also worth noting that in the AFM images, the structure of the control biofilms seemed more compact compared to those treated with peptide Hs02. These results corroborate those obtained by the live/dead staining.

### 2.3. Peptide Hs02 Decreased Bacterial Membrane Fluidity

The effect of peptide Hs02 on bacterial cytoplasmic membrane fluidity was assessed after calculating the Laurdan generalized polarization (GP). Laurdan is a polarity-sensitive fluorescent probe used to detect changes in the general membrane fluidity; an inverse relationship exists between Laurdan GP values and the degree of cytoplasmic membrane lipid order (i.e., lower GP values equate to greater membrane fluidity) [30,31,32].

As shown in Table 2, peptide Hs02 induced an increase in Laurdan GP values, reflecting a shift towards rigidification. This is in accordance with data that others [33] have obtained regarding the lipopeptide antibiotic daptomycin, which is active against Gram-positive pathogens. Daptomycin is a last-resort antibiotic for the treatment of infections caused by multidrug-resistant Gram-positive pathogens, such as MRSA, and is one of the few peptide antibiotics that can be administered intravenously [34]. Accordingly, others [35,36] have also reported that some AMPs can reduce the membrane fluidity of *S. aureus*. The antimicrobial hexapeptide MP196 and the cyclic β-sheet peptide gramicidin S also increased the Laurdan GP values in *S. aureus* and *Bacillus subtilis*, indicating that they also reduced the membrane fluidity [37].

The increase in Laurdan GP values was more marked in *P. aeruginosa* strains than in those of *S. aureus*. Moreover, as we can observe, membrane fluidity seems to be more variable among *S. aureus* strains than among *P. aeruginosa* strains. In the control condition, the Laurdan GP value for *S. aureus* ATCC 25923 was significantly lower than that obtained for the MRSA strains, Sa1 and Sa3, which was in accordance with previous reports [30].

## 3. Materials and Methods 

### 3.1. Hs02 Peptide

The peptide Hs02, primary structure KWAVRIIRKFIKGFIS-NH_2_, derived from the protein NP_001094891.3, was chemically synthesized using the Fmoc/t-butyl strategy [38] as described elsewhere [27]. For further details of the peptide purification and mass spectrometric analysis to confirm purity and primary structure, please refer to the literature [27].

### 3.2. Bacterial Strains and Growth Conditions

A range of susceptible and resistant strains were used in this study. Susceptible strains included *Pseudomonas aeruginosa* ATCC 27853, *P. aeruginosa* PAO1, *Escherichia coli* ATCC 25922, *Staphylococcus aureus* ATCC 25923, and *Enterococcus faecalis* ATCC 29212, as well as food isolates of *E. coli* (TBX1/1 and TBX2/3) and clinical isolates of *P. aeruginosa* (PA007 and PA008). Multidrug-resistant (MDR) clinical isolates of *P. aeruginosa* (Pa3, Pa4, PA002, PA004, and PA006) and of *S. aureus* (Sa1, Sa3, and SA007) were also used in this study. The antimicrobial resistance profile of all multidrug-resistant isolates is shown in Table A1 (see Appendix A). These bacteria were grown on Mueller–Hinton agar (MH; Liofilchem s.r.l., Roseto degli Abruzzi (Te), Italy) from stock cultures. MH plates were incubated at 37 °C for 20 h and then used to prepare fresh cultures for each experimental in vitro assay.

### 3.3. MIC and MBC Determination

The minimum inhibitory concentration (MIC) values of peptide Hs02 against the above-mentioned strains were determined by the broth microdilution method using cation-adjusted Mueller–Hinton broth (CAMHB; Sigma-Aldrich, Saint Louis, USA), and following the Clinical and Laboratory Standards Institute (CLSI) guidelines [39]. The minimum bactericidal concentration (MBC) was determined as reported by Bessa et al. [40].

### 3.4. Biofilm Formation Inhibition Assay

The effect of peptide Hs02 at concentrations equal to the MIC, 0.5× MIC, and 0.25× MIC on biofilm formation by isolates Sa1, SA007, PA004, and PA008 was assessed using the crystal violet assay as described previously [40]. Biofilms were formed in 96-well microtiter plates using tryptic soy broth (TSB; Liofilchem s.r.l., Roseto degli Abruzzi, Italy) and a starting inoculum of 1 × 10^6^ colony-forming units (CFU)/mL.

### 3.5. Performed Biofilm Treatment Assay

The effect of peptide Hs02 on 24-h established single-species biofilms of *P. aeruginosa* (Pa4-SA2, PA002, and PA004) and *S. aureus* (Sa1, Sa3, and SA007) and on 24-h dual-species biofilms (*P. aeruginosa* and *S. aureus*) was investigated. The dual-species combinations studied were: Pa3+Sa1, Pa4+SA007, and PA002+Sa3. Prior to choosing these combinations, a few other combinations were previously studied; nevertheless, in some of those mixed-culture biofilms grown for 24 h without treatment, *P. aeruginosa* clearly outgrew *S. aureus*, and therefore, such combinations were not selected. 

Briefly, biofilms were allowed to form for 24 h in 96-well microtiter plates, and then the planktonic phases were gently removed and the wells were rinsed and filled with concentrations of peptide Hs02 equal to 8× MIC. The OD_600_ was measured at time 0 h (just after the treatment with peptide Hs02) and time 24 h, after incubation for 24 h at 37 °C. Lower OD_600_ in the treated biofilms correlates to a reduction in the biofilm proliferation.

### 3.6. Visualization of Biofilms by CLSM

Single- and dual-species biofilms (Pa4 and SA007, PA002 and Sa3) grown for 24 h were formed on µ-Dishes 35 mm high, with ibidi polymer coverslips (ibidi GmbH, Germany), from a starting inoculum of 1 × 10^6^ CFU/mL in TSB. After 24 h, biofilms were rinsed with phosphate-buffered saline (PBS) and treated with a concentration of peptide Hs02 (8× MIC) for another 24 h. Control biofilms were formed in the same way but not treated with peptide. All biofilms were then rinsed and stained using the live/dead staining BacLight bacterial viability kit (Molecular Probes, Thermo Fisher Scientific, USA). Biofilms were examined by a laser scanning confocal system Leica TCS SP5 II (Leica Microsystems, Germany), equipped with (i) an inverted microscope, Leica DMI6000-CS, using a HC PL APO CS 63× /1.30 glycerin 21 °C objective and the lasers diode 405 nm and DPSS561 561 nm, and (ii) the LAS AF software. Two to three independent experiments were performed for CLSM visualization.

### 3.7. Visualization of Biofilms by AFM

The dual-species biofilm of PA002+Sa3 was grown for 24 h on Thermanox circular (15 mm diameter) plastic coverslips (Thermo Scientific, NY, USA) placed in 35 mm diameter polystyrene plates. After 24 h, biofilms were rinsed with PBS and treated with 128 µg/mL of peptide Hs02 and incubated for a further 24 h at 37 °C. The respective control biofilm was formed in the same way but in the absence of the peptide. All biofilms were rinsed three times with 1 mM of phosphate buffer and air-dried before AFM imaging. Samples were scanned with a TT-AFM from AFMWorkshop in air in vibrating mode. A 50 µm scanner and 300 kHz silicon cantilevers (ACT, AppNano) were used. Images were processed using Gwyddion 2.47 software. Two independent experiments were performed for AFM visualization.

### 3.8. Membrane Fluidity Assessment by Laurdan Generalized Polarization (GP)

The membrane fluidity of *P. aeruginosa* (ATCC 27852, PA002, and PA004) and *S. aureus* (ATCC 25923, Sa1, and Sa3) strains in the presence and absence of peptide Hs02 was determined by assessing the Laurdan generalized polarization (GP) as previously described [30,31] with some modifications. Briefly, fresh colonies were used to inoculate nutrient broth (NB; Liofilchem s.r.l., Roseto degli Abruzzi, Italy) to obtain cell suspensions with an OD_600_ of 0.4. Several aliquots of 1.5 mL of these bacterial suspensions were taken and centrifuged (9000 rpm, 8 min). For each strain, the bacterial pellets obtained were then resuspended in 1.5 mL of NB (in duplicate, to serve as controls; one to be unlabeled and the other labeled with Laurdan) and NB containing 0.5× MIC, MIC, or 2× MIC of peptide Hs02. These new bacterial suspensions were incubated at 37 °C for 3 h. Afterwards, they were centrifuged (9000 rpm, 8 min) and cells were washed twice in 15 mM Tris–HCl buffer (pH 7.4) and finally resuspended in 10 μM of Laurdan (from a 2 mM stock solution in dimethylformamide). Each suspension was incubated in the dark at 37 °C with shaking (500 rpm) for 1.5 h. Aliquots of 1 mL were transferred to a 1 cm quartz cuvette, and Laurdan emission spectra were obtained in a Varian Cary Eclipse fluorescence spectrofluorometer (Agilent Technologies, Santa Clara, California, USA) at an excitation wavelength of 350 nm using emission wavelengths from 410 to 550 nm. The temperature was set at 37.0 ± 0.1 °C. The excitation GP was calculated using the following equation: GP = (I_440_ − I_490_)/(I_440_ + I_490_)(1)
where I_440_ and I_490_ are fluorescence intensities at 440 and 490 nm, respectively. 

### 3.9. Statistical Analysis

The assay to assess the membrane fluidity by Laurdan generalized polarization was performed in three independent experiments, with the results being expressed as mean values ± standard deviation.

The biofilm formation and the preformed biofilm treatment assays were carried out in two independent experiments, with each experiment being performed in triplicate.

The results regarding the biofilm formation were expressed as mean values ± standard deviation. The statistical significance of differences between controls and experimental groups was evaluated using the Student’s *t*-test. *P*-values of < 0.05 were considered statistically significant.

## 4. Conclusions

By taking together the results from the MIC and MBC values and the live/dead staining, we can attribute a bactericidal action to the peptide Hs02, likely due to a direct effect on the bacterial cells by disrupting the cytoplasmic membrane. Moreover, the ability of the peptide to decrease the membrane fluidity of both *P. aeruginosa* and *S. aureus* strains also suggests a membrane-targeting antibacterial mechanism.

Peptide Hs02 hampered the proliferation and decreased the viability of single- and dual-species biofilms of two major pathogens, *P. aeruginosa* and *S. aureus*, confirming its potential as a lead for development towards an antibiofilm agent in complex cases involving polymicrobial biofilms.

## Figures and Tables

**Figure 1 ijms-20-03604-f001:**
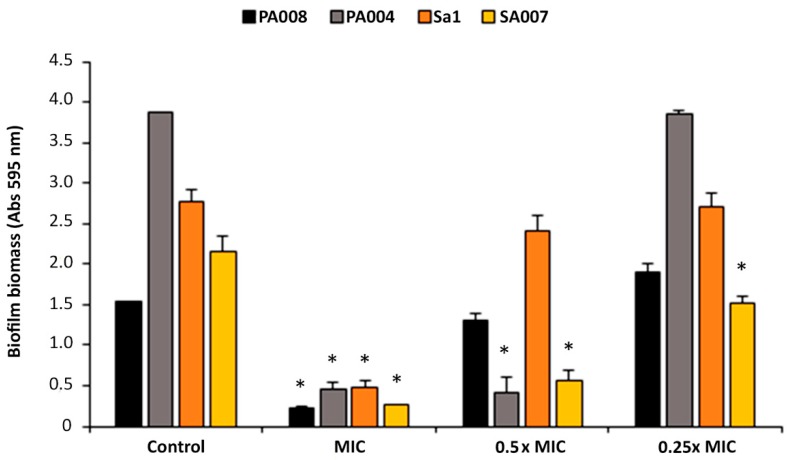
Biomass quantification through the crystal violet assay of biofilms formed by two *P. aeruginosa* and two methicillin-resistant *S. aureus* (MRSA) isolates in presence of peptide Hs02. Biofilms were formed in the presence of different concentrations (ranging from the MIC to 0.25× MIC) of peptide Hs02. Two independent experiments were performed in triplicate. Error bars represent SD. Statistically significant differences in comparison to the control (*p* < 0.05) are marked with an asterisk (*). Abs: Absorbance.

**Figure 2 ijms-20-03604-f002:**
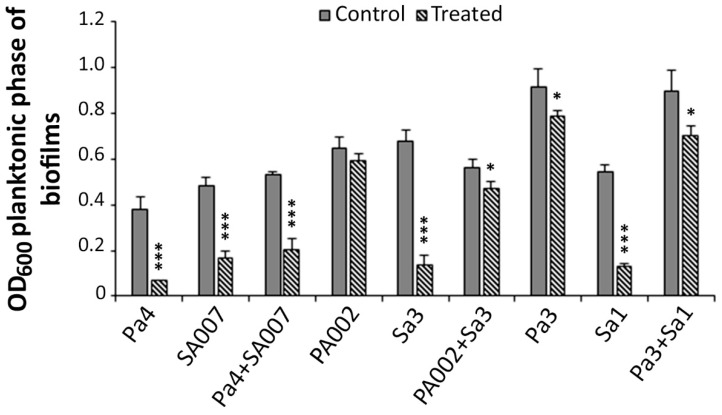
Effect of peptide Hs02 on the proliferation of 24-h preformed biofilms. The OD_600_ of planktonic phases of biofilms was measured and used to infer the biofilm proliferation. Pa4 and SA007 biofilms were treated with 8× MIC (32 µg/mL in both cases) of peptide Hs02; the dual-species biofilm of Pa4+SA007 was also treated with the same concentration. PA002 and Sa3 biofilms were treated with 8× MIC; that is, 128 µg/mL and 64 µg/mL, respectively, while the dual-species biofilm PA002+Sa3 was treated with 128 µg/mL. Pa3 and Sa1 biofilms were treated with 8× MIC; that is, 40 and 80 µg/mL, respectively, while the dual-species biofilm Pa3+Sa1 was treated with 80 µg/mL. Two independent experiments were performed in triplicate. Error bars represent SD. Statistically significant differences in comparison to the control are highlighted for *p* < 0.001 (***) or for 0.01 ≤ *p* < 0.05 (*). OD_600_: optical density at 600 nm.

**Figure 3 ijms-20-03604-f003:**
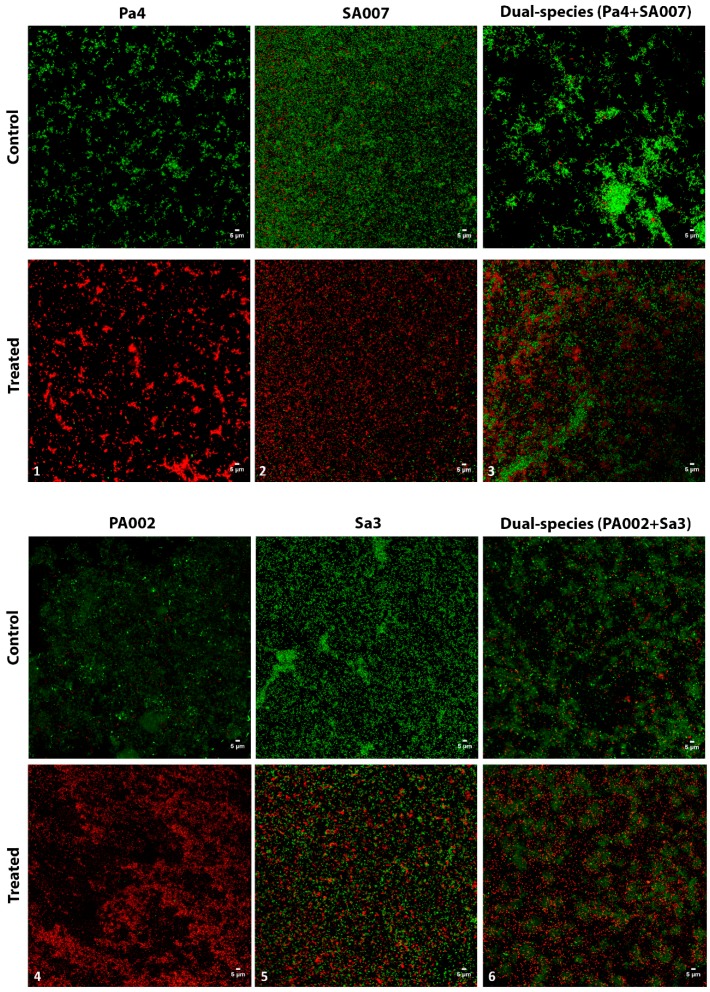
Confocal laser scanning microscopy (CLSM) images of single- and dual-species multidrug-resistant (MDR) *P. aeruginosa* and methicillin-resistant *S. aureus* (MRSA) biofilms. Biofilms were grown for 24 h and then treated for further 24 h with peptide Hs02 (**1**–**6**). Control images: no antimicrobial added. **1**, **2**, and **3**: 8× MIC peptide Hs02 (32 µg/mL); **4**: 8× MIC (128 µg/mL); **5**: 8× MIC (64 µg/mL). **6**: 128 µg/mL.

**Figure 4 ijms-20-03604-f004:**
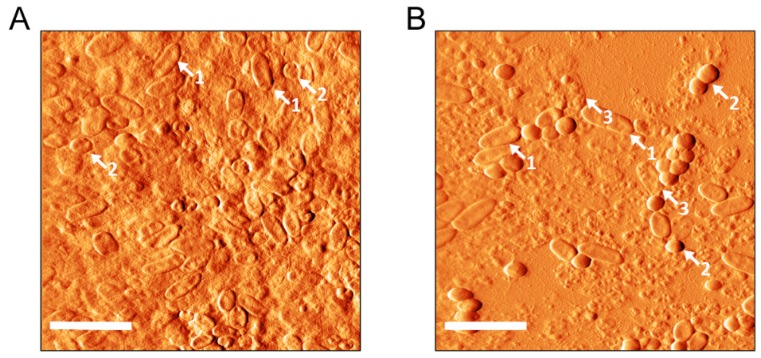
AFM images of the dual-species biofilm of PA002+Sa3. Biofilms were grown for 24 h and then treated for further 24 h with no peptide Hs02 (**A**) or with 128 µg/mL of the peptide (**B**). Scale bars correspond to 5 µm. Some features discussed in the text are indicated as follows: 1: undamaged PA002 cells; 2: undamaged Sa3 cells; 3: damaged PA002 cells.

**Table 1 ijms-20-03604-t001:** Minimum inhibitory concentration (MIC) and minimum bactericidal concentration (MBC) values of peptide Hs02 against several susceptible and multidrug-resistant strains.

Strains		MIC µg/mL (µM)	MBC µg/mL (µM)
**Reference strains**	***E. coli* ATCC 25922**	4 (2.0)	4 (2.0)
	***P. aeruginosa* ATCC 27853**	8 (4.1)	8 (4.1)
	***S. aureus* ATCC 25923**	8 (4.1)	8 (4.1)
	***E. faecalis* ATCC 29212**	16 (8.2)	16 (8.2)
***E. coli* strains**	***E. coli* TBX1/1** ^(S)^	4 (2.0)	4 (2.0)
	***E. coli* TBX2/3** ^(S)^	2 (1.0)	2 (1.0)
	**Ec1-SA1** ^(R)^	4 (2.0)	4 (2.0)
	**EC001** ^(R)^	4 (2.0)	4 (2.0)
***P. aeruginosa* strains**	**PAO1** ^(S)^	8 (4.1)	8 (4.1)
	**PA007** ^(S)^	8 (4.1)	8 (4.1)
	**PA008** ^(S)^	8 (4.1)	8 (4.1)
	**PA006** ^(R)^	4 (2.0)	4 (2.0)
	**Pa4** ^(R)^	4 (2.0)	4 (2.0)
	**PA002** ^(R)^	16 (8.2)	16 (8.2)
	**PA004** ^(R)^	8 (4.1)	8 (4.1)
	**Pa3** ^(R)^	4 (2.0)	4 (2.0)
***S. aureus* strains**	**Sa1** ^(R)^	8 (4.1)	8 (4.1)
	**SA007** ^(R)^	4 (2.0)	4 (2.0)
	**Sa3** ^(R)^	8 (4.1)	8 (4.1)
***E. faecalis* strain**	**Ef1** ^(R)^	4 (2.0)	4 (2.0)

^(S)^ Susceptible strain; ^(R)^ multidrug-resistant strain.

**Table 2 ijms-20-03604-t002:** Laurdan generalized polarization (GP) values obtained for different bacterial strains in the presence and absence of peptide Hs02 (in concentrations equal to 0.5× MIC, MIC, and 2× MIC). GP values allow the measurement of membrane fluidity.

	Control	0.5× MIC	MIC	2× MIC
***P. aeruginosa* ATCC 27853**	0.068 ± 0.002	0.077 ± 0.003 *	0.123 ± 0.004 *	0.155 ± 0.003 *
***S. aureus* ATCC 25923**	0.007 ± 0.000	0.008 ± 0.000 *	0.014 ± 0.002 *	0.038 ± 0.002 *
**PA002**	0.015 ± 0.001	0.060 ± 0.003 *	0.118 ± 0.003 *	0.151 ± 0.002 *
**PA004**	0.065 ± 0.002	0.122 ± 0.002 *	0.165 ± 0.001 *	0.205 ± 0.002 *
**Sa1**	0.088 ± 0.003	0.057 ± 0.030	0.115 ± 0.002 *	0.148 ± 0.002 *
**Sa3**	0.117 ± 0.001	0.110 ± 0.004 *	0.093 ± 0.003 *	0.123 ± 0.001 *

Statistically significant differences in comparison to the control (*p* < 0.05) are marked with an asterisk (*).

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
