# Peer review of "Intragenic Antimicrobial Peptide Hs02 Hampers the Proliferation of Single- and Dual-Species Biofilms of P. aeruginosa and S. aureus: A Promising Agent for Mitigation of Biofilm-Associated Infections"

_ijms, 2019, doi:10.3390/ijms20143604_

Reviewer 1 Report

1.    General comments

In the manuscript, the antimicrobial and antibiofilm activities of Hs02 were investigated against several P. aeruginosa and S. aureus strains.

Hs02 was bactericidal against all strains and MDR isolates. It also reduced the viability of the bacterial cells within the single and dual-species biofilms and induces a membrane rigidification in both P. aeruginosa and S. aureus. The result provides that Hs02 can potentially be a lead for further improvement as an antibiofilm agent.

2. Major revision

1) Figure 2 and line100~103

Although statistically significant differences in comparison to the control were marked as asterisks in Figure 1, but not in Figure 2. It is strongly recommended to draw statistically significant differences as asterisks in Figure 2 and reconsider the sentences of line 100~103 according to the revised Figure 2, including asterisks.

2) Table 2 

It is essential to draw statistically significant differences in comparison to the control as asterisks in Table 2.

3) Figure 4 and Line 130

It is recommended to show the positions of normal and damaged (in the form of empty membranes or “ghosts”) cells of PA002 and Sa3 strains in Figure 4, respectively.

3.    Minor revision

1a) Line 25~26: It is recommended to reconsider the sentence “Though Hs02 did not hamper the biofilm formation, it clearly” to “Hs02 hampered the biofilm formation in the presence of MIC concentration and clearly”.

1b) Line 84~88: It is recommended to reconsider the sentences of line 84~88 as in the following.

Peptide Hs02 did not inhibited biofilm formation by the PA004, PA008, and Sa1 and SA007 isolates in the presence of MIC concentrations (Figure 1), because in the presence of MIC concentrations, the biofilm formed was similar or greater than the control biofilm (grown in absence of peptide Hs02). Nonetheless, For isolate SA007, at sub-MICs of 1/2× and 1/4× MIC, the biofilm formed was significantly reduced in comparison to the control biofilm (grown in absence of peptide Hs02), and for isolate PA004, the biofilm formed was also reduced at 1/2×MIC.

2) Line 168~169: As it remains unclear whether the literature [28] is accepted or not, it is recommended to show the details concerning the peptide purification and mass spectrometric analysis as an Appendix or Supplementary Files".

Author Response

We would like to thank Reviewer 1 for his comments, which will allow us to improve our manuscript.

Below, we provide a point-by-point response to the reviewer’s comments.

Reviewer: 1

2. Major revision

1) Figure 2 and line100~103

Although statistically significant differences in comparison to the control were marked as asterisks in Figure 1, but not in Figure 2. It is strongly recommended to draw statistically significant differences as asterisks in Figure 2 and reconsider the sentences of line 100~103 according to the revised Figure 2, including asterisks.

Authors’ response: As suggested, statistically significant differences in comparison to the control were inserted as asterisks in Figure 2. The mentioned sentence was revised appropriately.

2) Table 2 

It is essential to draw statistically significant differences in comparison to the control as asterisks in Table 2.

 Authors’ response: Statistically significant differences were highlighted with an asterisk.

3) Figure 4 and Line 130

It is recommended to show the positions of normal and damaged (in the form of empty membranes or “ghosts”) cells of PA002 and Sa3 strains in Figure 4, respectively.

Authors’ response: The requested modifications were performed in Figure 4 and respective figure’s legend.

3.    Minor revision

1a) Line 25~26: It is recommended to reconsider the sentence “Though Hs02 did not hamper the biofilm formation, it clearly” to “Hs02 hampered the biofilm formation in the presence of MIC concentration and clearly”.

Authors’ response: No formation of biofilm in presence of the MIC is expected since the peptide inhibits the bacterial growth (and in this case is even bactericidal). The ability of a compound to inhibit the biofilm formation is assessed when it is present at sub-inhibitory concentrations. Only if at sub-MICs it can inhibit or reduce the biofilm formed in respect to the control, we can say it inhibits/hampers the biofilm formation. Therefore, we have rewritten the sentence as follows: “Though Hs02 was not able to hamper the biofilm formation by some strains at sub-MICs, it clearly affected 24-h preformed biofilms”.

1b) Line 84~88: It is recommended to reconsider the sentences of line 84~88 as in the following.

Peptide Hs02 did not inhibited biofilm formation by the PA004, PA008, and Sa1 and SA007 isolates in the presence of MIC concentrations (Figure 1), because in the presence of MIC concentrations, the biofilm formed was similar or greater than the control biofilm (grown in absence of peptide Hs02). Nonetheless, For isolate SA007, at sub-MICs of 1/2× and 1/4× MIC, the biofilm formed was significantly reduced in comparison to the control biofilm (grown in absence of peptide Hs02), and for isolate PA004, the biofilm formed was also reduced at 1/2×MIC.

Authors’ response: That is not quite accurate. We disagree with the reviewer correction which is contradicting the effective results: Peptide Hs02 did not inhibited biofilm formation by the PA004, PA008, and Sa1 and SA007 isolates in the presence of MIC concentrations (Figure 1), because in the presence of MIC concentrations, the biofilm formed was similar or greater than was significantly lower than the control biofilm (grown in absence of peptide Hs02).

Therefore, we have added one more sentence and adjusted accordingly the others, however, keeping the same interpretation of the results, which can be observed in Figure 1.

2) Line 168~169: As it remains unclear whether the literature [28] is accepted or not, it is recommended to show the details concerning the peptide purification and mass spectrometric analysis as an Appendix or Supplementary Files".

Authors’ response: In what regards the fact that reference [28] was not yet accepted for publication, we have contacted the Managing Editor dealing with our manuscript and he replied: “Actually, reference part could be revised before paper publication, or even before the issue releasing. So I would encourage you to resubmit it back for further peer reviewing. We could change this reference during final proofreading if this paper could be accepted.”

We also know that reference [28] is now very likely to be accepted by Plos One, since the authors have received a first feedback as “Minor revision requested”, and therefore, they are also proceeding to submit their revision to Plos One.

Reviewer 2 Report

A well written research paper with high significance for the microbiology field.

I would suggest the publication of the manuscript after taking into consideration the following minor corrections

1- In figure 1 (Biomass quantification) the author should specify how many replicates the standard error bars represents.

2- Line 91, a reference for the crystal violet procedure should be mentioned.

3- Reference 28 is unpublished, so should be replaced with another one to confirm purity of the primary structure and  peptide purification.

Author Response

We would like to thank Reviewer 2 for his comments, which will allow us to improve our manuscript.

Below, we provide a point-by-point response to the reviewer’s comments.

Reviewer: 2

1- In figure 1 (Biomass quantification) the author should specify how many replicates the standard error bars represents.

Authors’ response: That information, which is stated in the methods, was inserted in Figure 1’s legend as well as to Figure 2’s.

2- Line 91, a reference for the crystal violet procedure should be mentioned.

Authors’ response: Such reference is mentioned in the methods section when the method is described, therefore, we do not think it is necessary to introduce it in the figure’s legend as the reviewer suggests.

3- Reference 28 is unpublished, so should be replaced with another one to confirm purity of the primary structure and peptide purification.

Authors’ response: We know that reference [28] is now very likely to be accepted for publication by Plos One, since the authors have received a first feedback as “Minor revision requested”, and therefore, they are also proceeding to submit their revision to Plos One. On that matter, and since we would like to insert the reference [28] as a paper already published, we have contacted the Managing Editor of IJMS dealing with our manuscript, and his reply was: “Actually, reference part could be revised before paper publication, or even before the issue releasing. So I would encourage you to resubmit it back for further peer reviewing. We could change this reference during final proofreading if this paper could be accepted.” Therefore, we are pretty sure we will convert reference [28] to a definite published paper before our present manuscript be published.

Round  2

Reviewer 1 Report

The manuscript is properly revised according to my comments.